# SENTKVCOMPRESS: SENTENCE-LEVEL DYNAMIC KVCACHE COMPRESSION FOR EFFICIENT LONG-CONTEXT LLM INFERENCE

## ABSTRACT

The demand for million-token-scale contexts in the agent era has expanded the KVCache to terabyte levels, creating severe inference bottlenecks due to high storage overhead and frequent memory access. Existing KV compression methods struggle to alleviate this memory pressure efficiently, facing challenges with the accuracy or efficiency of KVCache storage and usage. **Challenge 1–At the KV storage level**, KV preprocessing mechanisms face accuracy and efficiency challenges: those with information loss suffer from a significant end-to-end accuracy drop of over 30%, while near-lossless methods incur substantial overhead, resulting in a superlinear increase in inference time with context length. **Challenge 2–At the usage level**, KV selection mechanisms face a similar dilemma. Static Selection (e.g., Attention Sink) fails to capture semantic relationships, yielding low recall (<50% for top-10 tokens). Dynamic selection (e.g., online score calculation) incurs prohibitive overhead, consuming over 60% of KV selection GPU time and 70% of CPU-GPU memory bandwidth with redundant transfers.

**Our core insight** is that the above challenges arise because existing methods follow an unstructured, token-level compression paradigm. This focus on discrete tokens, which inherently lack semantic structure, forces the model to expend substantial additional computation to implicitly re-extract structural information from long texts during inference. To address this, we observe that attention scores aggregate naturally at the sentence level. Leveraging this finding, we propose SentKVCompress, a novel sentence-level dynamic KV cache management framework that explicitly extracts and utilizes this inherent structural information. **At the KVCache storage level**, to address the challenge of accuracy loss and high preprocessing overhead, we propose a Sentence-level perceived KVCache preprocess framework, maintaining accuracy while cutting overhead to below 20%. **At the KVCache usage level**, to address the challenge of imprecise selection and high additional overhead, we propose a sentence semantic-driven KVCache selection strategy, enabling 70% of KVs to be reused. Experiments show that SentKVCompress achieves a maximum speedup of 4.2× with nearly no accuracy loss, and reduces the peak memory by 2.7× in long context scenarios, while also achieving the highest accuracy at equivalent KV usage rates. The code will be open-source in https://github.com/Indexleaf475/ICLR26-SentKVCompress

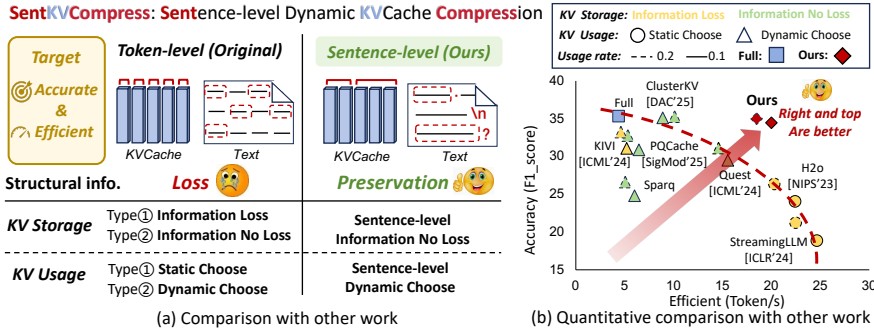

Figure 1: (a) The biggest difference between our work and others: sentence-level. (b) Compared to existing methods, our method achieves a better "accuracy-efficient" trade-off.

## 1 INTRODUCTION

KVCache is critical for large language model (LLM) inference: it stores historical key-value vectors, cuts redundant attention computations, and reduces computational load. In the Agent era, widespread long-context tasks (e.g., deep research OpenAI (2025); Anthropic (2025) and code generation (Qwen Team, 2025; Anthropic, 2025)) boost demand for long-text inference. LLM providers thus enhance models' long-context capabilities, advancing to million-token-level processing (e.g., Claude 3.7 (Anthropic, 2025), Gemini 2.5 Pro (Comanici et al., 2025)).

The growing context volume leads to a steady increase in the storage space occupied by KVCache. The KVCache is a tensor that is positively correlated with text length. When the sequence length reaches 512K, at the KVCache storage level, even for models based on the GQA (Ainslie et al., 2023) mechanism, the memory consumption still reaches 80GB (equivalent to the memory capacity of high-end GPUs such as NVIDIA A100 (NVIDIA Corporation, 2020) and NVIDIA H100 (NVIDIA Corporation, 2022)). At the KVCache usage level, the latency caused by frequent memory access during attention computation also increases accordingly—the time spent on loading KVCache accounts for more than 80% of the total time when LLMs generate tokens. This memory bottleneck poses a severe challenge to inference efficiency.

In this paper, we first analyze the limitations of current related work in the context of the trend of increasing context length, and then address the question:

*Amid the trend of increasingly long context, how to design an efficient long context inference method for LLMs that also balances response quality?*

Prior works have proposed a variety of efficient KVCache methods (Xiao et al., 2023; Li et al., 2024; Liu et al., 2024; Tang et al., 2024; Zhang et al., 2024), which can be summarized by the classification system shown in Table 1 (Appendix A for details). This represents a summary of technical methodologies from two key perspectives: KV storage and KV usage. However, the current methods still face the following challenges in moving toward million-token-scale long-text inference scenarios:

***Challenge-1**: At the KV storage level, KV preprocessing mechanisms have limitations in terms of accuracy or efficiency.* As shown in Figure 2, KV preprocessing mechanisms of information loss (e.g., Evicting (Xiao et al., 2023), Merging (Li et al., 2024)) involve permanent pruning of information in one or more dimensions (sequence length, head dimension, etc.) of KVCache, leading to unavoidable loss of important semantic information in long-text scenarios and resulting in an accuracy drop of over 30%. For KV preprocessing mechanisms of information non-loss (e.g., Clustering (Liu et al., 2024), Low-rank Decomposition (Chang et al., 2025)), the computational complexity exhibits a superlinear growth relationship with text length (e.g., approaching $O(L^2)$), which causes superlinear growth in time. Additionally, in terms of accuracy, they do not support ultra-long context scenarios that require pagination. Compared with one-time clustering, split clustering reduces the accuracy of ClusterKV (Liu et al., 2024) from 57% to 32%.

***Challenge-2: At the KVCache usage level, the important KV selection mechanisms have limitations in terms of accuracy or efficiency.*** The static selection mechanisms (e.g., Historical attn. score (Zhang et al., 2023), Attn. sink (Xiao et al., 2023)) often rely on statistical patterns and fail to capture semantic relationships. In long-text scenarios, the recall rate of top-10 tokens in terms of attention scores drops below 50%. For instance, with a context length of 10K and a token budget of 256, the recall rates of H2o (Zhang et al., 2023) and StreamingLLM (Xiao et al., 2023) are 47

Table 1: Related Work Summary

| Classification | | KV Usage | |
|---|---|---|---|
| | | **Static Selection** | **Dynamic Selection** |
| **KV Storage** | **Information loss** | KV Evicting 
 High efficiency, Low accuracy | KV Quantization 
 Medium efficiency, Medium accuracy |
| | **Information near lossless** | None[*] | KV loading 
 Low efficiency, High accuracy |

[*] The combination of static selection with near-lossless information preservation leads to model accuracy degradation, hence no existing work adopts this approach.

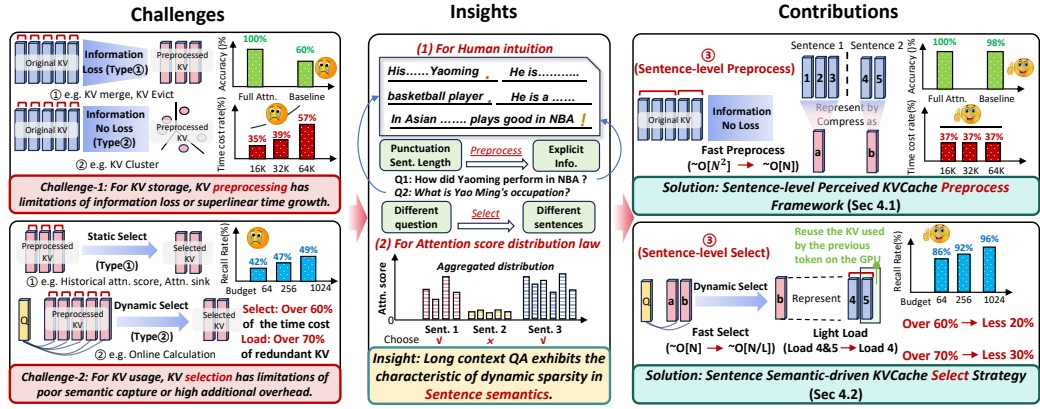

Figure 2: Overview. (a) Challenges: The upper and lower sections introduce the challenges in KV storage and KV usage. (b) Insights: Switching from token-level to sentence-level. (c) Contribution: Two solutions proposed by us address the aspects of KV storage and KV usage.

**Our core insight is**: The root cause of the above challenges is that existing methods follow an unstructured token-level compression approach. This approach overlooks the inherent semantic structure of text, thereby forcing the model to consume substantial additional computation during inference to re-extract structured information from long texts. Based on the observations and experiments illustrated in Figure 2 "Insight", we propose an explanatory paradigm that the transition **from token-level to sentence-level holds great potential**.

Our contributions, as shown in Figure 2, are summarized as follows:

***Contribution-1: We propose the insight of approaching KVCache compression at the sentence-level.*** We find that when using sentence-level KVCache compression on public datasets, even with a KV usage rate of 10%, the accuracy score is improved compared to full attention, increasing from 26.27 to 26.86. Based on this insight, we design our method from the perspective of sentence-level KV storage and usage.

***Contribution-2: At the KVCache storage level, to address the issues of information loss and high additional overhead in preprocessing, we propose the Sentence-level Perceived KVCache Preprocess Framework.*** We design a sentence segmentation algorithm based on punctuation information and distance information for preprocessing, reducing the KV preprocessing overhead to the $O(L)$ level. We also design a supporting sentence compression algorithm to reduce the sentence storage space while preserving sentence information, such that the memory required for important KV preselection is only 7% of the original KV memory.

***Contribution-3: At the KVCache usage level, to address the issues of inaccurate selection and high additional overhead, we propose the Sentence-level Perceived KVCache Utilization Framework.*** We design a sentence selection method with a variable-length block processing strategy, transforming KV selection from the token-level to the sentence-level, which reduces the overhead of important KV selection on GPUs to 20%. Furthermore, to reduce redundant communication overhead, we design a KV reuse algorithm integrated with variable-length sentence block processing, enabling 70% of KVs to be reused.

## 2 PROBLEM SETTING

### 2.1 PROBLEM SETTING

The computational flow of LLMs inference mainly consists of the attention process and the Feed-Forward Network (FFN) process. The KVCache is utilized in the attention process. As the length of the input text $L$ increases, the attention process becomes the primary bottleneck in terms of time overhead (over 90%), see Table 4, 3 in the Appendix for detailed data.

The attention mechanism computes scores using query matrix $Q \in \mathbb{R}^{n \times d}$, key matrix $K \in \mathbb{R}^{n \times d}$, and value matrix $V \in \mathbb{R}^{n \times d}$. Its output $O$ is typically calculated as:

$$O = \text{softmax}\left(\frac{QK^T}{\sqrt{d}}\right) V \qquad (1)$$

**KV Storage**: Define compression ratio $C = \frac{S_{\text{processed}}}{S_{\text{original}}}$, where $S_{\text{original}}$ and $S_{\text{processed}}$ are the original and processed storage sizes of $K/V$ matrices, respectively. A smaller $C$ indicates higher compression.

**KV Usage**: Define usage rate $U = \frac{N_{\text{transferred}}}{N_{\text{total}}}$, where $N_{\text{transferred}}$ is the number of original $K/V$ elements involved in attention data transfer, and $N_{\text{total}}$ is the total original elements. A lower $U$ reduces transfer overhead (boosting inference speed); retrieving most high-attention $K$ under low $U$ ensures response accuracy.

As text length $n$ grows:

1. **Computational Complexity**: The $QK^T$ matrix multiplication in attention has complexity $O(n^2 d)$. As $n$ increases, this cost grows quadratically, significantly raising attention time.

2. **KVCache Transfer**: Transferring KVCache from CPU to GPU introduces major overhead $T_{\text{transfer}}$. Since KVCache size is proportional to $n$, and $T_{\text{transfer}} = \frac{S_{\text{KVCache}}}{B}$ (where $B$ = CPU-GPU bandwidth, $S_{\text{KVCache}}$ = KVCache size), larger $n$ increases $S_{\text{KVCache}}$ and thus $T_{\text{transfer}}$, further raising attention overhead.

A central challenge in long-context LLM inference lies in the trade-off between efficiency and accuracy. Although techniques like aggressive KV cache compression (low $C$) or restricted usage (low $U$) can address the escalating computational and memory overheads, they risk severe accuracy loss due to information discarding. Striking a balance between these dimensions is therefore critical.

## 3 INSIGHT

Beyond our intuitive understanding, we have also conducted validity verification to explore the sentence-level KVCache paradigm.

**Intuition: Inspiration from the Migration of Human Text Processing Modes.** Humans naturally dissect long texts sentence by sentence and aggregate semantics when understanding them. Instead of memorizing word by word, they grasp the overall meaning efficiently by extracting sentence-level semantic chunks (*e.g.*, basketball player, "Asian representative", "NBA performance").

**Theory: Complexity Advantage of Sentence-Level Processing.** From the perspective of computational complexity, the storage and selection overhead of token-level KVCache (Key-Value Cache) grows superlinearly with text length $L$ (e.g., the complexity of clustering methods is close to $O(L^2)$). However, sentence-level processing aggregates the granularity from tokens to sentences (assuming each sentence contains $M$ tokens on average, and the number of sentences is $L/M$), which reduces the complexity to the order of $O(L)$.

**Experiment: Sentence-Level Aggregation of Attention Score Distribution.** The model's attention focus on long texts exhibits a sentence-block aggregation feature. Important semantics are concentrated in sentence-level units rather than scattered across individual tokens. Informed by the observation that sentence-level KV cache compression at a 10% usage rate outperforms full attention (26.86 vs. 26.27) on public datasets, we designed our method around sentence-level storage and usage.

## 4 SENTKVCOMPRESS

### 4.1 SENTENCE-LEVEL PERCEIVED KVCACHE PREPROCESS FRAMEWORK

To address the core limitations of token-level KVCache storage—specifically information loss and excessive preprocessing overhead—this study proposes a **Sentence-Level Perceived KVCache Preprocessing Framework**. This framework leverages the sentence-level aggregation property of

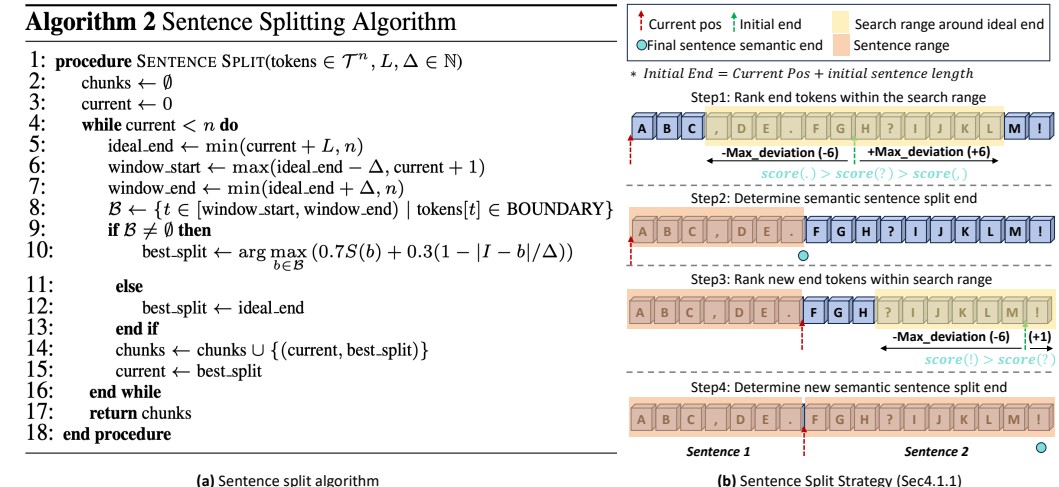

**(a)** Sentence split algorithm        **(b)** Sentence Split Strategy (Sec4.1.1)

Figure 3: (a) Pseudocode for Sentence Split Execution. (b) Illustration of Alg.1: Sentence Split Strategy Based on Punctuation and Length Information.

attention scores in long-context language models (LLMs) and achieves efficient, accurate KVCache preprocessing through three sequential steps. Its overall design adheres to an $O(n)$ time complexity, ensuring scalability for long texts at the million-token scale.

### 4.1.1 SENTENCE SPLIT STRATEGY BASED ON PUNCTUATION AND LENGTH INFORMATION

To split long texts into manageable sentence chunks suitable for model processing, we propose a semantic splitting algorithm (Figure 3 (a) Algo.) that leverages both punctuation cues and length constraints. The primary objective is to avoid fragmenting coherent semantic units, thereby ensuring that each segmented chunk is both length-appropriate and semantically coherent.

The core of our strategy involves identifying optimal split points within a sliding window around a target length. Specifically, for a current starting index *current*, the algorithm first calculates an ideal endpoint *ideal_end = current + L*, where $L$ is a predefined target length. To allow flexibility, a search window of size $\pm\Delta$ around *ideal_end* (i.e., $[ideal\_end - \Delta, ideal\_end + \Delta]$) is defined.

Within this window, the algorithm identifies all predefined semantic boundary tokens ($\mathcal{B}$), such as periods (.), question marks (?), and exclamation marks (!).(Appendix Figure 9 for Detail) The optimal split point *best_split* is selected from these candidates by maximizing a scoring function that balances semantic importance and length adherence:

$$\text{Score}(b) = 0.7 \cdot S(b) + 0.3 \cdot \left(1 - \frac{|ideal\_end - b|}{\Delta}\right)$$

Here, $S(b)$ denotes the semantic weight of the boundary token at position $b$, prioritizing stronger sentence delimiters. The second term penalizes deviations from the ideal length $L$. If no boundary token is found within the search window, the algorithm defaults to splitting at *ideal_end*. As illustrated in Figure 3, the process iterates: after selecting the best split point for the current segment, it becomes the new starting point for the next segment. This iterative process continues until the entire text is processed, resulting in a list of $(start, end)$ index pairs. This method effectively balances the dual requirements of semantic integrity and length uniformity, producing high-quality input chunks for downstream tasks.

### 4.1.2 SENTENCE COMPRESS STRATEGY

The Sentence Compression Strategy maintains, for each sentence, a dynamic profile of extreme key vector values to support subsequent cache selection. During inference, as each new sentence's key vector $\mathbf{k} \in \mathbb{R}^{\dim}$ is added, two global statistic vectors—the maximum key vector $\mathbf{M}$ and the minimum key vector $\mathbf{m}$—are updated in an element-wise fashion. For each dimension $i$, the update

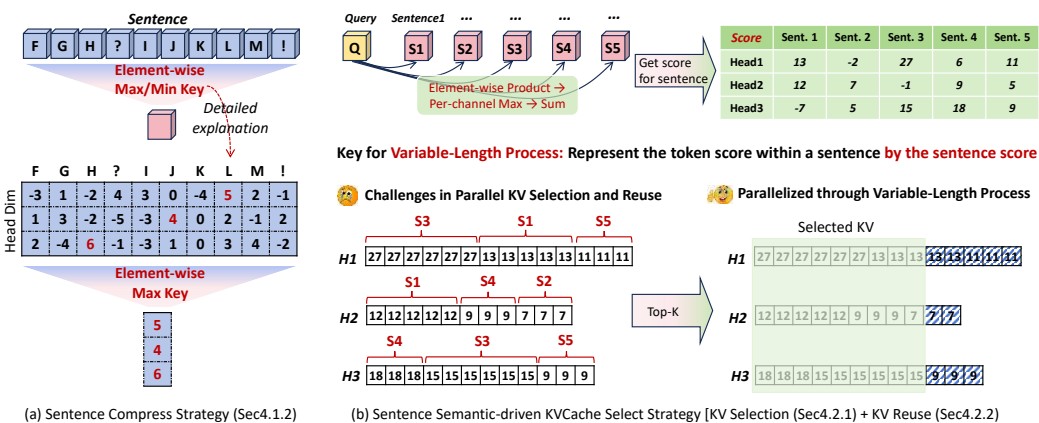

Figure 4: (a) Sentence Compress Strategy. (b) Sentence Semantic-driven KVCache Select Strategy.

rules are:

$$\mathbf{M}_i \leftarrow \max(\mathbf{M}_i, \mathbf{k}_i), \quad \mathbf{m}_i \leftarrow \min(\mathbf{m}_i, \mathbf{k}_i).$$

This process continuously tracks the value range $[\mathbf{m}_i, \mathbf{M}_i]$ for each feature dimension across all cached keys. The resulting ranges provide a normalized basis for assessing token importance, forming the foundation of the compression strategy, as shown in Figure 4 (a).

## 4.2 SENTENCE SEMANTIC-DRIVEN KVCACHE SELECT STRATEGY

To enhance the semantic capability and reduce the overhead of KVCache selection, we propose a sentence semantics-driven selection strategy, incorporating both important sentence selection and KV reuse. However, due to varying sentence lengths, its application leads to variable-length blocks, which hinders parallel computation. Therefore, we have specifically designed a **variable-length block processing strategy** as a complementary solution, as shown in Figure 4.

### 4.2.1 SENTENCE SELECTION STRATEGY WITH VARIABLE-LENGTH BLOCK PROCESSING

**The aforementioned sentence segmentation strategy divides the input text into independent semantic chunks based on semantic integrity (rather than fixed-length segments)**. Each chunk naturally corresponds to a complete sentence or semantic unit, which inherently preserves sentence-level semantics while allowing the chunk length to be dynamically adjusted according to the content of the sentences. For each variable-length semantic chunk, sentence importance scores are generated through attention score calculation, which we then map to token importance scores—we assume that the importance score of a token can be represented by the sentence it belongs to. Finally, a top-k operation (where the k value is dynamically controlled by the token budget) is applied to filter out the indices of core tokens. This process does not rely on the original length of the chunk. Instead, it dynamically intercepts effective tokens based on semantic importance, ensuring that chunks of varying lengths all converge to a unified token budget after filtering.

### 4.2.2 KV REUSE STRATEGY WITH VARIABLE-LENGTH BLOCK PROCESSING

**To address the variable-length block issue and enable KV reuse, our approach proceeds as follows.** When processing variable-length blocks during KV reuse, we first iterate through each attention head. For each head, we compute the reusable KV. Next, across multiple heads, we truncate the reusable KV based on the minimum reusable data volume, which ensures a consistent length of reusable KV caches among all heads, overcoming the inconsistency caused by variable-length blocks. Subsequently, we load the required new KV and the truncated reusable KV in parallel. Finally, we merge the data from all heads (including both the new KV and the truncated reusable KV) to complete the KV reuse process, enabling efficient utilization of cached KV information even with variable-length blocks. As shown in the Figure 4 (b), after the lengths are made the same, we can perform parallel reuse. The specific reuse steps are provided in the Appendix A.

## 5 EXPERIMENT

In this section, we conduct extensive experiments to answer the following research questions (**RQs**):

- **RQ1**: How does SentKVCompress perform in terms of effectiveness in long context QA?
- **RQ2**: How is the efficiency of SentKVCompress?
- **RQ3**: What does each component of SentKVCompress bring?

### 5.1 EXPERIEMNTAL SETTINGS

#### 5.1.1 DATASETS AND METRICS

**Datasets.** We adopt **LongBench** (Bai et al., 2023), a well-known benchmark for long-context scenarios for the evaluation of our work. LongBench comprises 21 datasets spanning 6 task categories in both English and Chinese. These tasks cover key long-text application areas, including single-document QA, multi-document QA, summarization, few-shot learning, and so on. As the base models we use are predominantly pre-trained on English data, we focus on the English datasets within LongBench for our assessment. The data length distribution in LongBench spans 0-8K+ tokens, enabling comprehensive and precise evaluation of LLMs' long-context understanding capabilities.

**Metrics.** The inference accuracy and inference efficiency will be the key metrics to be evaluated. For the inference accuracy, we will continue to employ the **F1 score** in LongBench (Bai et al., 2023) to measure the similarity between the model outputs and the ground-truth answers. For the inference efficiency, we will test the Time to Generate First Token (**TTFT**), the Time per Output Token (**TPOT**), and the end-to-end latency (**E2e**).

#### 5.1.2 BASELINES AND IMPLEMENTATION DETAILS

**Baselines.** To comprehensively and fairly evaluate the effectiveness and efficiency of SentKVCompress, we select several types of representative state-of-the-art retrieval methods as our baselines.

- **Methods for token-level fixed choice** mainly includes **KV Evicting**. We choose StreamingLLM (Xiao et al., 2023), the first work that proposes the concept of attention sink. We also choose H2o (Zhang et al., 2023), which propose Heavy Hitter Oracle (H2O), a KV cache eviction policy.
- **Methods for token-level dynamic choice** mainly includes **KV Loading** (We have selected Quest (Tang et al., 2024), ClusterKV (Liu et al., 2024), PQCache (Zhang et al., 2024).) **KV Quantization** (We choose KIVI).

**Implementation details.** We used Mistral-7B-inst-v0.2 GQA model (32K), Longchat-32K, and Deepseek R1 Distill Llama8B as the foundational models. All of our experiments are carried out on NVIDIA A800 80GB GPU, the initial sentence length was set to 14, and Max deviation was set to 8. All baselines were implemented in accordance with the descriptions in their respective papers.

**Support for advanced foundational LLMs.** To verify the support of our method for advanced foundational LLMs, we also adopted DeepSeek-R1-Distill-Llama-8B (DeepSeek-AI, 2025) as the foundational model. The experimental results are shown in Figure 5. The results indicate that with a usage rate of 0.1, our method has already achieved performance close to that of full attention.

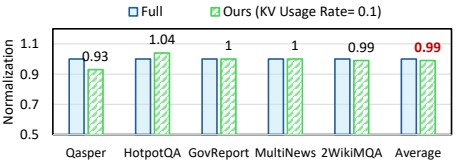

Figure 5: Support for Deepseek R1 Distill Llama8B

### 5.2 EFFECTIVENESS ANALYSIS (RQ1)

**Accuracy.** To verify the response quality of our method, we conducted experiments on LongBench. We comprehensively tested the performance of our method and the baselines under the same KV

Table 2: Main accuracy results. In the three categories of related work, bold indicates the first rank in performance, while underlining indicates the second rank in performance.

| KV Usage Rate = 0.1 | Method | NarrativeQA | HotpotQA | 2WikiMQA | GovReport | Triviaqa | Average |
|---|---|---|---|---|---|---|---|
| KV Evicting | H2o | 19.08 | 23.8 | 18.13 | 23.81 | 83.79 | 33.72 |
| | StreamingLLM | 20.56 | 22.25 | 17.38 | 20.26 | 85.06 | 33.1 |
| KV Quantization | KIVI(0.125) | 25.76 | 41.33 | 25.34 | 32.32 | 86.19 | 42.19 |
| KV Loading | PQCache | 20.65 | 30.87 | 21.79 | 30.87 | 86.12 | 38.06 |
| | Quest | 24.06 | 32.9 | 18.65 | 30.67 | 78.47 | 36.95 |
| | ClusterKV | **25.79** | 41.14 | 25.45 | **33.22** | **86.53** | 42.43 |
| | Sparq | 21.54 | 24.8 | 17.76 | 24.8 | 86.15 | 35.01 |
| | **Ours** | 25.47 | **42.01** | **26.26** | 33.15 | 86.2 | **42.62** |
| Full KV | Full | 26.47 | 43.72 | 26.97 | 32.59 | 85.74 | 43.10 |

| | StreamingLLM | H2o | Sparq | ClusterKV | Quest | PQCache | KIVI | Full | Ours |
|---|---|---|---|---|---|---|---|---|---|
| E2e (Tokens/s) | 24.69 | 22.48 | 5.06 | 10.1 | 16.56 | 5.33 | 4.52 | 4.41 | 18.5 |

(a) E2e overview

TTFT (s) — TPOT (s)

PQCache: 1.55 / 0.92 ClusterKV: 2.9 / 0.12 Quest: 1.32 / 0.1 Sparq: 1.69 / 0.92 Ours: 1.32 / 0.093

(b) TTFT&TPOT Analysis about KV Loading

Figure 6: Efficiency Analysis

usage rates, and the experimental results are presented in Table 2. Results show that SentKVCompress achieves higher average accuracy than all mainstream baselines, surpassing even methods like ClusterKV that require substantial preprocessing overhead, demonstrating its significant potential as a novel and effective paradigm.

## 5.3 EFFICIENCY ANALYSIS (RQ2)

**GPU Inference Efficiency Analysis.** Time to Generate First Token (TTFT) measures the time to generate the First Token. Time Per Output Token (TPOT) measures the time of each decoding step. End-to-End Latency (E2e) measures the time of a complete inference. As shown in the Figure 6, we measured the E2e latency of various methods across different datasets. For one of the datasets, we conducted a detailed analysis of all KV Loading methods that belong to the same method category as our proposed method. Works such as ClusterKV and PQCache, which require clustering during the prefill stage and even cluster updates in the decoding stage, may suffer from low prefill or decoding efficiency due to process blocking. Owing to the sequential computation and communication of SPARQ, SPARQ exhibits relatively high latency, despite having negligible additional computational overhead. In contrast, our method involves no additional computational overhead similar to clustering; moreover, it addresses communication overhead through a variable-length block processing strategy, thus achieving the best performance across all efficiency metrics. KV Quantization like KIVI show limited advantages in single-batch scenarios, as they primarily focus on reducing inference memory, with their main benefits being realized in multi-batch settings.

**CPU-GPU Inference Efficiency Analysis.** We also implemented the KV offload operation for our method to verify its acceleration effect in ultra-long context scenarios—where KVCache continuously grows and exceeds GPU memory capacity as the context length increases. The experimental results are shown in Figure 7. They demonstrate that in the CPU-GPU scenario, with comparable accuracy, our method achieves a maximum speedup of 5.9x over the full attention baseline.

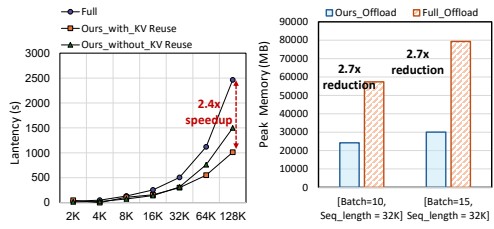

Figure 7: CPU-GPU Inference Efficiency Analysis.

## 5.4 ABLATION ANALYSIS (RQ3)

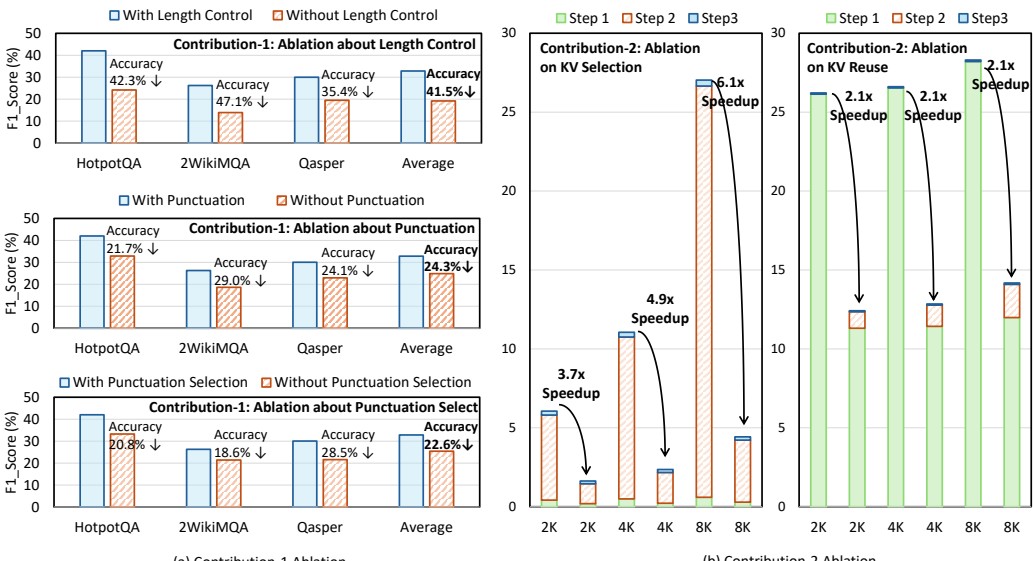

Figure 8: Ablation Study.

**Contribution-1 Ablation.** As shown in Figure 8 (a), our core argument is that sentence semantic split should be based on two key informational cues: length control and punctuation marks. To illustrate the necessity of this argument, we conducted ablation experiments from three perspectives: *without length control*, *without punctuation*, and *without punctuation selection*. The results show that the absence of any one of these three components leads to an average accuracy drop of more than 22%. This indicates that the information to be considered for sentence splitting should not be limited to merely punctuation-based splitting or length control. The specific steps for Step 1, Step 2, and Step 3 are provided in the Appendix A. The additional overhead here occurs in Step 1 and accounts for less than 50% of Step 1's total cost; nonetheless, it yields significant end-to-end acceleration.

**Contribution-2 Ablation.** As shown in Figure 8 (b), our core argument is that the design of a variable-length processing strategy is crucial for sentence-level important KV selection and KV reuse. To demonstrate the necessity of this argument, we conducted ablation experiments from two aspects: important KV selection and KV reuse. Experiments show that after adopting the variable-length processing strategy designed by us, the speedup ranges from 2.1x to 6.1x, which plays a significant role in long-context inference. The specific steps for Step 1, Step 2, and Step 3 are provided in the Appendix A. The additional overhead introduced here stems from Step 2; however, the experimental results show that while this additional overhead accounts for a very small proportion, it achieves significant overall acceleration.

## 6 CONCLUSION

In this work, we analyzed existing approaches from the perspectives of KV storage and usage, proposing a novel paradigm for KVCache management: **sentence-level**. Based on this paradigm, we conducted designs from the perspectives of KV storage and usage, focusing on two key strategies: sentence split based on punctuation and length information, and sentence semantic-driven KVCache selection. Experiments show that SentKVCompress achieves the highest accuracy under the same KV usage rates. It yields average speedups of 4.2x and 2.4x in GPU-only and CPU-GPU scenarios, respectively, and achieves a 2.7x reduction in peak memory usage in CPU-GPU offload scenarios. These results strongly demonstrate the potential of sentence-level KV. Future work will explore system-algorithm co-design based on this concept.

ETHICS STATEMENT

All authors of this paper have read and strictly adhere to the ICLR Code of Ethics. We confirm that our research does not involve human subjects, sensitive data (e.g., personal, medical, or financial information), or applications with potential harmful impacts on individuals or society. We remain committed to the responsible stewardship of research as outlined in the ICLR Code of Ethics, ensuring our work advances knowledge in a manner consistent with public good and well-being.

REPRODUCIBILITY STATEMENT

All findings presented in this paper are fully reproducible. We will provide anonymized code. Detailed information about our experiments, including hyperparameters, training protocols, and evaluation methods, can be found in the "Experiments" section. We are confident that, with the provided resources, readers will be able to reproduce all of the results presented in this paper.

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

## A  APPENDIX

### USE OF LLMS

In accordance with the ACM Policy on Authorship, this disclosure states that during the preparation of this manuscript, generative artificial intelligence (GenAI) tools were solely used for polishing the authors' original text. Specifically, such tools were employed to optimize linguistic expressions to align with academic norms and the stylistic requirements of research papers, including tasks like grammatical proofreading and sentence structure adjustment (i.e., light editing). Notably, these tools were not utilized to generate new content, such as paragraphs, figures, data, or core arguments, all of which represent the authors' independent intellectual contributions.

### SUPPLEMENTARY MATERIALS ON PROBLEM SETTING

Table 3: Time Cost Ratios of SDPA Attention on Nvidia A800

| Context | Device | Attn. Type | Attn./FFN | FFN/Total |
|---------|--------|-----------|-----------|-----------|
| 1k | Nvidia A800 | sdpa | 3.96 | 5.00% |
| 2k | Nvidia A800 | sdpa | 3.86 | 4.00% |
| 3k | Nvidia A800 | sdpa | 3.62 | 3.78% |
| 4k | Nvidia A800 | sdpa | 3.55 | 3.30% |
| 5k | Nvidia A800 | sdpa | 4.8 | 2.74% |
| 6k | Nvidia A800 | sdpa | 4.74 | 2.61% |
| 7k | Nvidia A800 | sdpa | 4.74 | 2.42% |
| 8k | Nvidia A800 | sdpa | 4.82 | 2.32% |
| 9k | Nvidia A800 | sdpa | 4.86 | 2.22% |
| 10k | Nvidia A800 | sdpa | 4.8 | 2.00% |

For Table 3: In the GPU scenario, regarding the analysis of the overhead proportion of Attention (Attn.) and FFN layers when context grows, the execution time of Attention will become the primary bottleneck as large models scale.

For table 4, in the CPU-GPU scenario, due to the limited PCIe bandwidth between the CPU and GPU, the transfer time overhead of moving useful KVCache from the CPU (storage unit) to the GPU (computation unit) during attention computation becomes the primary bottleneck.

### SUPPLEMENTARY MATERIALS ON RELATED WORK

**Token-level fixed choice mainly consists of related works such as KV Evicting and KV Merging.** They (e.g., StreamingLLM (Xiao et al., 2023), H2o (Zhang et al., 2023), SnapKV (Li et al., 2024)) define a fixed compression and usage rule by exploring the distribution patterns of attention scores of KVCache across sequence length, attention heads, model layers, and attention dimensions. At the KV storage level, it permanently discards some useless vectors through techniques such as eviction and merging to achieve the goal of reducing KVCache memory, with a compression ratio typically ranging from 12% to 43%. At the KV usage level, it statically selects important KVs using historical attention scores or attention sink patterns for use in attention computation. The compression and usage overhead of such works is relatively low (O[1]), thus enabling fast inference speed, as shown in Figure 2(a).

**Token-level dynamic choice mainly consists of related works such as KV Loading and KV Quantization.** KV Loading (e.g., Quest (Tang et al., 2024), ClusterKV (Liu et al., 2024), PQ-Cache (Zhang et al., 2024)) defines a dynamic compression and usage rule by preprocessing KVs and exploring the relationships between query vectors and KV vectors. At the KV storage level, these methods usually do not perform explicit compression; instead, they conduct implicit semantic processing on KVs through techniques like clustering and low-rank decomposition (Sun et al., 2024), with a compression ratio typically ranging from 58% to 105%. At the KV usage level, they

Table 4: KV Transfer Time, GPU Computation Time, and Transfer Time Proportion Under Different Context Lengths

| Context | KV transfer time [CPU-GPU] (ms) | GPU computation time (ms) | Proportion of transfer time |
|---|---|---|---|
| 50 | 0.69 | 0.42 | 0.621621622 |
| 100 | 1.12 | 0.2 | 0.848484848 |
| 200 | 2.27 | 0.39 | 0.853383459 |
| 500 | 5.35 | 0.38 | 0.933682373 |
| 1k | 9.3 | 0.64 | 0.935613682 |
| 2k | 16.61 | 0.67 | 0.961226852 |
| 3k | 25.15 | 0.76 | 0.970667696 |
| 4k | 30.81 | 0.81 | 0.974383302 |
| 5k | 37.6 | 0.7 | 0.981979629 |
| 6k | 45.94 | 0.86 | 0.981623932 |
| 7k | 52.88 | 0.9 | 0.983265154 |
| 8k | 60.27 | 0.82 | 0.986577181 |
| 9k | 67.45 | 0.93 | 0.986399532 |
| 10k | 75.83 | 0.85 | 0.988914971 |
| 16k | 104.74 | 0.86 | 0.991856061 |
| 32k | 208.95 | 1.16 | 0.994526416 |
| 50k | 365.54 | 1.35 | 0.99634758 |
| 64k | 520.84 | 1.94 | 0.99628907 |
| 128k | 918.78 | 2.16 | 0.99765457 |

dynamically select important KVs for each token generation by calculating the importance scores between the query (Q) and all KVs in real time. This method achieves a relatively high recall rate of important KVs, hence ensuring high accuracy, as shown in Figure 2(b). KV Quantization (e.g., KIVI, KVQuant) uses information loss approaches at the KV storage level, pruning KV tensors via quantization (e.g., converting the original 8-bit precision format to 4-bit or 2-bit), while retaining dynamic selection at the usage level.

SUPPLEMENTARY MATERIALS ON METHOD ABOUT PUNCTUATION

In our experiment, we explored the impact of using different quantities of punctuation marks on sentence compression performance in various models. We selected a different number of punctuation marks for each model, considering factors such as vocabulary size, release date, and architectural characteristics to optimize performance in the sentence compression task. For LongChat-v1.5-7b-32k (Kwon et al., 2023), with a smaller vocabulary size (32,000) and an earlier release date (August 2023), we chose the most basic single punctuation mark to simplify the task and enhance stability. For DeepSeek-R1-Distill-Llama-8B (DeepSeek-AI, 2025), with a larger vocabulary size (128,256) and a more recent release date (January 2025), we selected the majority of common punctuation marks to improve compression performance. For Mistral-7B-Instruct-v0.2 (Jiang et al., 2023), with a vocabulary size of 32,768 and a moderately recent release date (March 2024), we chose an intermediate number of punctuation marks to balance the model's processing capability and task requirements. The specific punctuation marks selected for each model, along with their corresponding token IDs and weights, are shown in Figure 9.

SUPPLEMENTARY MATERIALS ON METHOD ABOUT KV SELECTION AND KV REUSE

**To compute the attention scores and perform KV cache reuse, our approach proceeds as follows:**

1. **Compute attention scores for each semantic chunk:** Based on the attention values between each semantic chunk and the query, we calculate the attention scores in the shape

of $[B, H, Q = 1, S]$. Then, we apply a top-k operation to obtain the top-k token indices (kvcache indices) in the shape of $[B, H, Q = 1, \text{token\_budget}]$, where $k$ is dynamically controlled by the token budget.

2. **Parallel loading of required KV cache:** We then parallelly load only the required KV cache based on the top-k token indices.

3. **Query @ required Key and calculate attention:** For the query and the required key, we compute the attention weights in the shape of $[B, H, Q = 1, \text{token\_budget}]$, apply softmax to the attention weights, and finally perform the attention operation between the attention weights and the required values to obtain the output.

**To address the variable-length block issue and enable KV reuse, our approach proceeds as follows:**

1. **Iterate through each attention head:** We compute the reusable KV for each head serially. Across multiple heads, we truncate the reusable KV based on the minimum reusable data volume. This ensures a consistent length of reusable KV caches among all heads, overcoming the inconsistency caused by variable-length blocks.

2. **Parallel loading of KV caches:** We load the required new KV and the truncated reusable KV in parallel. The truncated excess data from the reusable KV is combined with the new required data, forming a consistent length of the new required KV.

3. **Merge the data from all heads:** We combine the data from all heads (including both the new KV and the truncated reusable KV) to complete the KV reuse process. This allows efficient utilization of cached KV information, even with variable-length blocks.

As shown in Figure 4 (b), once the lengths are aligned, parallel reuse can be performed.

SUPPLEMENTARY MATERIALS ON EXPERIMENT

**Average KVCache Reuse Rate.**
We also implemented the KV offload operation for our method to verify its acceleration effect in ultra-long context scenarios—where KVCache continuously grows and exceeds GPU memory capacity as the context length increases. The experimental results are shown in Figure 10. They demonstrate that in the CPU-GPU scenario, with comparable accuracy, our method achieves a maximum speedup of 5.9x over the full attention baseline.

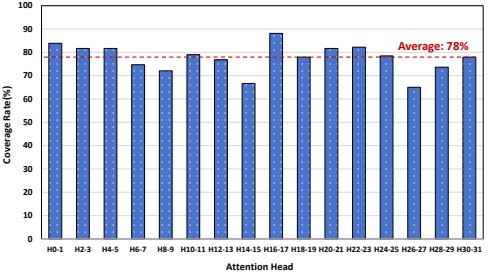

Figure 10: CPU-GPU Inference Efficiency Analysis.

```
longchat-v1.5-              Mistral-7B-               DeepSeek-R1-
   7b-32k                  Instruct-v0.2             Distill-Llama-8B

PUNCTUATION_WEIGHTS = {     PUNCTUATION_WEIGHTS = {    PUNCTUATIONS_WEIGHTS = {
    29889: 1.0,   # .           28723: 1.0,   # .         13   : 1.0,   # .
    29973: 0.9,   # ?           611  : 1.0,   # ."        382  : 1.0,   # .\n\n
    29991: 0.9,   # !           609  : 1.0,   # ).        627  : 1.0,   # .\n
    13   : 0.8,   # \n          2586 : 1.0,   # ".        570  : 1.0,   # ).
    2056 : 0.7,   # ;           842  : 1.0,   # .         1210 : 1.0,   # ."
    584  : 0.7,   # :           1101 : 1.0,   # ...       2266 : 1.0,   # ."\n\n
    29892: 0.6,   # ,           3850 : 1.0,   # ...       3343 : 1.0,   # ".
    539  : 0.5,   # "           28804: 0.9,   # ?         2637 : 1.0,   # .,
    29915: 0.5,   # '           1110 : 0.9,   # ?"        1131 : 1.0,   # ...
    29898: 0.5,   # (           28808: 0.9,   # !         4286 : 1.0,   # .\n\n\n
    29897: 0.5,   # )           2781 : 0.9,   # !"        4390 : 1.0,   # ).\n
    518  : 0.5,   # [           13   : 0.8,   # \n        6266 : 1.0,   # .)
    519  : 0.5,   # ]           28745: 0.7,   # ;         2195 : 1.0,   # ...\n\n
}                              344  : 0.7,   # );        9456 : 1.0,   # .)\n\n
                               2753 : 0.7,   # ;         3677 : 1.0,   # ).\n\n
                               28747: 0.7,   # :         18976: 1.0,   # .:
                               6210 : 0.7,   # ::        75223: 1.0,   # .\n\n\n\n\n
                               12813: 0.7,   # .:        29275: 1.0,   # .)\n
                               714  : 0.7,   # :         10246: 1.0,   # ."\n
                               28725: 0.6,   # ,         11690: 1.0,   # ".\n\n
                               557  : 0.6,   # ),        3238 : 1.0,   # .'
                               862  : 0.6,   # ,"        4527 : 1.0,   # '.
                               1200 : 0.6,   # ,         30   : 0.9,   # ?
                               548  : 0.6,   # ",        1980 : 0.9,   # ?\n\n
                               28742: 0.5,   # '         0    : 0.9,   # !
                               464  : 0.5,   # '         5380 : 0.9,   # ?\n
                               1815 : 0.5,   # .'        2268 : 0.9,   # !\n\n
                               28732: 0.5,   # (         12241: 0.9,   # ?"\n\n
                               325  : 0.5,   # (         7673 : 0.9,   # ?"
                               28731: 0.5,   # )         9135 : 0.9,   # !"
                               1143 : 0.5,   # )         4999 : 0.9,   # !\n
                               28792: 0.5,   # [         17642: 0.9,   # !"\n\n
                               733  : 0.5,   # [         198  : 0.8,   # \n
                               28793: 0.5,   # ]         340  : 0.8,   # )\n
                               1181 : 0.5,   # ]         512  : 0.8,   # :\n
}                                                        345  : 0.8,   # ,\n
                                                         271  : 0.8,   # \n\n
                                                         ......
                                                     }
```

Figure 9: The actual punctuation marks and weight conditions used in different models.

