# OpenReview forum: "SentKVCompress: Sentence-Level Dynamic KVCache Compression for Efficient Long-Context LLM Inference"
_ICLR.cc/2026/Conference — ICLR 2026 Conference Withdrawn Submission_

### Official Review · Reviewer_Xhab · 2025-10-16

**Soundness:** 3
**Presentation:** 1
**Contribution:** 2
**Rating:** 4
**Confidence:** 4

**Summary:**

This paper introduces SentKVCompress, a sentence-level approximate attention that is integrated with KV cache offloading. SentKVCompress mainly introduces a sentence segmentation algorithm, followed by sentence KV cache compression technique, and a KV cache reuse (caching) technique when offloading is applied. SentKVCompress achieves notable speedups, reaching up to 4.2x acceleration with 2.7x peak memory reduction, without significant performance degredation. This paper benchmarks SentKVCompress on LongBench, compared with classical baselines, such as H2O, StreamingLLM, etc., where SentKVCompress surpasses all the baselines.

**Strengths:**

1. Segmenting the input context at the sentence level is both efficient and effective, as selecting different sentences leads to better task performance. The overall design of the paper is reasonable.

2. The benchmark results show a significant improvement in inference efficiency without performance degradation. SentKVCompress demonstrates strong effectiveness.

**Weaknesses:**

1. Limited novelty. KV cache offloading and eviction have been widely studied. There are existing methods that perform KV selection at the sentence level, such as SentenceKV [1]. The paper should clarify the major difference between this work and SentenceKV.

2. The benchmark datasets used are rather limited. The test entries in LongBench are relatively short and easy; therefore, more challenging benchmarks such as RULER [2] and LongBench-v2 [3] should be included for evaluation.

3. More baselines should be discussed or included, such as SnapKV [4] and ShadowKV [5], to better position the proposed method within existing literature.

4. The overall writing should be improved. The current description of the method is vague and confusing. For example, in line 315, the term “KV reuse” appears before being clearly defined. All technical terms should be clearly introduced before use.

---

[1] Zhu, Yuxuan, et al. "SentenceKV: Efficient LLM Inference via Sentence-Level Semantic KV Caching." arXiv preprint arXiv:2504.00970 (2025).

[2] Hsieh, Cheng-Ping, et al. "RULER: What's the Real Context Size of Your Long-Context Language Models?." arXiv preprint arXiv:2404.06654 (2024).

[3] Bai, Yushi, et al. "Longbench v2: Towards deeper understanding and reasoning on realistic long-context multitasks." arXiv preprint arXiv:2412.15204 (2024).

[4] Li, Yuhong, et al. "Snapkv: Llm knows what you are looking for before generation." Advances in Neural Information Processing Systems 37 (2024): 22947-22970.

[5] Sun, Hanshi, et al. "Shadowkv: Kv cache in shadows for high-throughput long-context llm inference." arXiv preprint arXiv:2410.21465 (2024).

**Questions:**

See above.

---

> ### Author Response · Authors · 2025-11-30
>
> In response to your comment, we have addressed the lack of comparison with block-sparse attention baseline as follows:
>
> 1.Qualitative comparison
>
> Works like XAttention[1] primarily address the compute-bound problem, which enables their application beyond text to domains like video. In contrast, the memory-bound issue is more critical in long-text scenarios, which is the primary problem we aim to solve. For this reason, we have summarized relevant block-sparse works below and begin with a qualitative comparison. The core approach of InfLLM[2] and Quest[3] involves partitioning the historical KV Cache into fixed-length blocks and compressing each one. During the decode stage, the importance between the Query and each block is computed to pre-select and load critical blocks, thereby reducing the loading overhead for the subsequent QKV computation in the attention phase. The general approach of SentenceKV[4] is similar to the ones mentioned above. The key difference lies in its block split strategy, which splits the text based on punctuation into sentences. This method can better capture textual semantics and improve accuracy. However, the resulting variable-length blocks, due to varying sentence lengths, may impair parallel computing efficiency. Our work incorporates considerations for both the accuracy of semantic split&Length control and the computational speed of handling variable-length blocks.
>
> | Related block-sparse attention work | Semantic split | Variable-length block handling strategy | Block Length control |
> | :--- | :--: | ---: | ---: |
> | InfLLM[2] | ×   | × | √ |
> | Quest[3] | ×   | × | √ |
> | Sentencekv[4] | √   | × | × |
> | Ours | √   | √ | √ |
>
> 2.Quantitative comparison
>
> A comparison with the Quest [3] work has been previously mentioned in the original text. Additionally, we have benchmarked our method against the most relevant work, SentenceKV [4], to highlight our superior speed under comparable accuracy.
>
> | Accuracy and Latency | Triviaqa |  | 2Wikimqa |  | Qasper |  | Result Average |  |
> |:--------:|:--------:|:--:|:--------:|:--:|:--------:|:--:|:--------:|:--:|
> | Metric | E2e Latency(sec.) | Accuracy(F1 score) | E2e Latency(sec.) | Accuracy(F1 score) | E2e Latency(sec.) | Accuracy(F1 score) | E2e Latency(sec.) | Accuracy(F1 score) |
> | Sentencelkv | 31.07 | 74.89 | 23.75 | 22.64 | 26.09 | 29.08 | 26.97 | 42.20 |
> | Ours | 5.49 | 83.10 | 3.65 | 21.87 | 4.05 | 29.59 | 4.40 | 44.85 |
> | **Comparison** | **5.7×** | **+8.21** | **6.5×** | **-0.77** | **6.4×** | **+0.51** | **6.1×** | **+2.65** |
>
>
> **The most significant difference between our approach and sentencekv lies in the length control strategy during text split, punctuation choice during text split and the variable-length block processing strategy during sentence selection. This brings advantages in terms of precision and latency respectively. Next, I will provide a detailed explanation through ablation experiments.**
>
> | Accuracy:F1 Score | Triviaqa |  | 2Wikimqa |  | Qasper |  | Average |  |
> |:--------:|:--------:|:--:|:--------:|:--:|:--------:|:--:|:--------:|:--:|
> | Metric | E2e Latency(sec.) | Accuracy(F1 score) | E2e Latency(sec.) | Accuracy(F1 score) | E2e Latency(sec.) | Accuracy(F1 score) | E2e Latency(sec.) | Accuracy(F1 score) |
> | Sentencelkv | 31.07 | 74.89 | 23.75 | 22.64 | 26.09 | 29.08 | 26.97 | 42.20 |
> | ours | 5.49 | 83.10 | 3.65 | 21.87 | 4.05 | 29.59 | 4.40 | 44.85 |
> | **Comparison** | **5.7×** | **+8.21** | **6.5×** | **-0.77** | **6.4×** | **+0.51** | **6.1×** | **+2.65** |
>
> (1) Ablation on length control strategy
>
> | Accuracy: F1 score | Hotpotqa | GovReport | 2WikiMQA | Qasper |
> |--------------------|----------|-----------|----------|--------|
> | Without control    | 24.23    | 30.63     | 13.89    | 19.5   |
> | With control       | 42.01    | 33.15     | 26.26    | 30.2   |
> | **Comparison**         | **+17.78**   | **+2.52**     | **+12.37**   | **+10.7**  |
>
>
> (2) Ablation on punctuation choice strategy
>
> | Accuracy: F1 score | Hotpotqa | GovReport | 2WikiMQA | Qasper |
> |--------------------|----------|-----------|----------|--------|
> | Without choice     | 33.24    | 30.2      | 21.38    | 21.6   |
> | With choice        | 42.01    | 33.15     | 26.26    | 30.2   |
> | **Comparison**         | **+8.77**    | **+2.95**     | **+4.88**    | **+8.6**   |
>
>
> (3) Ablation on variable-length block processing strategy
>
> | Select Latency: ms | Context: 2K | Context: 4K | Context: 8K | Context: 16K | Context: 32K |
> |--------------------|-------------|-------------|-------------|--------------|--------------|
> | Without process    | 6.06        | 11.05       | 27.03       | 53.21        | 105.11       |
> | With process       | 1.63        | 2.36        | 4.43        | 7.50         | 13.75        |
> | **Comparison**         | **3.7x**        | **4.7x**        | **6.1x**        | **7.1x**         | **7.6x**         |
>
>
> All of the ablation experiments mentioned above are included in the current version of the article.

---

### Official Review · Reviewer_hX5P · 2025-10-29

**Soundness:** 2
**Presentation:** 2
**Contribution:** 2
**Rating:** 2
**Confidence:** 5

**Summary:**

The paper proposes SentKVCompress, a sentence-level framework for KV cache preprocessing and usage in long-context LLM inference. It first splits text into sentence-like chunks using a scoring rule that balances punctuation strength and a target length within a window, giving an O(n) splitting pipeline. This aims to keep semantic units intact while keeping lengths regular.

The method then computes sentence scores from attention-derived signals and maps them to token scores so that KV selection happens at the sentence level. To support reuse across steps and heads, it introduces a variable-length block processing strategy that aligns reusable KV across attention heads.

**Strengths:**

1. Sentence-aware splitting balances punctuation and target length, which reduces fragmentation and redundant KV while staying linear time. This is simple and practical.
2. Sentence-level selection and reuse reduce on-the-fly scoring and bandwidth pressure; the variable-length block alignment across heads is a useful engineering design.
3. The framework is modular and could be combined with other storage-side methods.

**Weaknesses:**

1. Missing very-long-context benchmarks. There is no RULER-style evaluation at 32k–64k and 128k where retrieval and reuse matter most. This is essential to validate scalability and accuracy retention at realistic agentic workloads.
2. Missing sentence-level baselines. A direct comparison with sentence-level offloading or caching methods such as SentenceKV[1] would test whether the gains come from sentence granularity itself or other design choices.
3. Limited analysis of the sentence-score construction. The paper maintains per-dimension max–min statistics over keys to normalize and score, but it does not test alternatives such as mean-pooled keys or other low-cost sketches; the choice seems heuristic.
4. Robustness to punctuation-sparse inputs is unclear. For math, code, or log-like text without strong punctuation, it is not clear how the splitter behaves and whether accuracy degrades.

[1] SentenceKV: Efficient LLM Inference via Sentence-Level Semantic KV Caching.    https://openreview.net/pdf?id=HyPeYU9JR6

**Questions:**

1. Why use per-dimension extreme values for key statistics rather than mean or robust mean as the sentence representation or normalization basis. Have you tried mean-pooled key vectors per sentence or per bucket?
2. What is the average and variance of the chunk length after your split on your main datasets, and how sensitive are results to the target length L and window.
3. How do you split texts for math or code tasks where punctuation is sparse or non-standard.

---

> ### Author Response · Authors · 2025-11-30
>
> Hello reviewer, thank you for your comments.
>
> We understand your comments as pertaining to two main aspects: first, the distinctions from related work, and second, matters of detail.
>
> ### 1.The distinctions from related works
>
> As for your comment "Additional baselines for related sentence-level segmentation approaches should be added.": We have selected Sentencekv[1], a representative work at the sentence level, to illustrate the differences. **The most significant difference between our approach and theirs lies in the length control strategy during text split, punctuation choice during text split and the variable-length block processing strategy during sentence selection.** This brings advantages in terms of precision and latency respectively. Next, I will provide a detailed explanation through ablation experiments.
>
> | Accuracy:F1 Score | Triviaqa |  | 2Wikimqa |  | Qasper |  | Average |  |
> |:--------:|:--------:|:--:|:--------:|:--:|:--------:|:--:|:--------:|:--:|
> | Metric | E2e Latency(sec.) | Accuracy(F1 score) | E2e Latency(sec.) | Accuracy(F1 score) | E2e Latency(sec.) | Accuracy(F1 score) | E2e Latency(sec.) | Accuracy(F1 score) |
> | Sentencelkv | 31.07 | 74.89 | 23.75 | 22.64 | 26.09 | 29.08 | 26.97 | 42.20 |
> | ours | 5.49 | 83.10 | 3.65 | 21.87 | 4.05 | 29.59 | 4.40 | 44.85 |
> | **Comparison** | **5.7×** | **+8.21** | **6.5×** | **-0.77** | **6.4×** | **+0.51** | **6.1×** | **+2.65** |
>
> (1) Ablation on length control strategy
>
> | Accuracy: F1 score | Hotpotqa | GovReport | 2WikiMQA | Qasper |
> |--------------------|----------|-----------|----------|--------|
> | Without control    | 24.23    | 30.63     | 13.89    | 19.5   |
> | With control       | 42.01    | 33.15     | 26.26    | 30.2   |
> | **Comparison**         | **+17.78**   | **+2.52**     | **+12.37**   | **+10.7**  |
>
>
> (2) Ablation on punctuation choice strategy
>
> | Accuracy: F1 score | Hotpotqa | GovReport | 2WikiMQA | Qasper |
> |--------------------|----------|-----------|----------|--------|
> | Without choice     | 33.24    | 30.2      | 21.38    | 21.6   |
> | With choice        | 42.01    | 33.15     | 26.26    | 30.2   |
> | **Comparison**         | **+8.77**    | **+2.95**     | **+4.88**    | **+8.6**   |
>
>
> (3) Ablation on variable-length block processing strategy
>
> | Select Latency: ms | Context: 2K | Context: 4K | Context: 8K | Context: 16K | Context: 32K |
> |--------------------|-------------|-------------|-------------|--------------|--------------|
> | Without process    | 6.06        | 11.05       | 27.03       | 53.21        | 105.11       |
> | With process       | 1.63        | 2.36        | 4.43        | 7.50         | 13.75        |
> | **Comparison**         | **3.7x**        | **4.7x**        | **6.1x**        | **7.1x**         | **7.6x**         |
>
>
> All of the ablation experiments mentioned above are included in the current version of the article.
>
> [1] Zhu Y, Falahati A, Yang D H, et al. SentenceKV: Efficient LLM Inference via Sentence-Level Semantic KV Caching[J]. arXiv preprint arXiv:2504.00970, 2025.
>
>
>
> ### 2.Matters of detail
> (1) As for your comment "the average and variance of the chunk length after your split": With a predefined length of 14, we calculated the mean and variance across different datasets as well as the non-predefined length rate. Based on this, we concluded that approximately 70% of the information originates from sentence semantics, while 30% comes from textual length and structural information. The average length of sentence blocks remains stable.
>
>
> |                         | HotpotQA | 2WikiMQA | Qasper |
> |-------------------------|----------|----------|--------|
> | **Average**             | 14.62    | 14.54    | 14.55  |
> | **Variance**            | 12.88    | 11.86    | 11.89  |
> | **Non-Predefine length rate** | 68.3%   | 69.3%   | 68.9% |
>
>
>
> (2) As for your comment "Why use per-dimension extreme values for key statistics": We used the mean-pooled method, but compared with the per-dimension max-min approach, it results in a drop in accuracy. This makes our framework more suitable for methods like per max/min, which can blend information from various tokens.
>
> | Accuracy and Latency | HotpotQA |
> |--------------------|----------|
> | Mean               | 15.36    |
> | Per max/min        | 32.67    |
>
>
> (3) As for your comment "How do you split texts for math or code tasks": We use symbols such as "/n", "(", and ")" as delimiters, and the rest of the framework can be executed as is.

---

### Official Review · Reviewer_7Vi3 · 2025-10-31

**Soundness:** 2
**Presentation:** 2
**Contribution:** 2
**Rating:** 2
**Confidence:** 4

**Summary:**

The paper introduces SentKVCompress, a sentence-level KV cache compression framework for efficient long-context LLM inference. The key idea is to leverage sentence-level semantic structure rather than token-level processing. The framework includes: 1. a sentence-level preprocessing stage that splits tokens based on punctuation/length heuristics and maintains per-sentence compressed statistics, and 2. a sentence semantic-guided KV selection scheme with variable length block processing. Experiments on LongBench report improved average performance and efficiency compared to baseline approaches.

**Strengths:**

1. Problem motivation is clear and motivated to try and capture semantic information to enhance KV cache compression methods. Length-control for semantic boundary splitting offer some noticeable improvements as seen from ablation studies.

2. Ablation studies provide evidence that both length control and punctuation cues matters.

3. Strong performance on QA tasks from LongBench, while being more efficient compared to KV loading baseline approaches tested.

**Weaknesses:**

1. The main idea of punctuation, sentence-level chunking is already explored in related work (e.g. SentenceKV [1]). The main difference here appears to be a length-control term in the splitter's scoring, which feels incremental in absence of a direct comparison. The paper should explicitly position against these works.

2. Presentation and clarity of the methodology needs to be improved. Some important details are left omitted from Section 4, leaving the reader to fill in the gaps themselves. For instance, Sec. 4.1.2 defines per-dim min/max key vectors but does not specify how these are actually use in the selection computation.

3. Justification for using per-head iteration versus aggregate scoring should be made clear. For re-use, Sec. 4.2.2 iterates per-head and truncates to align variable length blocks, which breaks the parallelism. Some explanation for why per-head choices is theoretically or empirically better is needed to demonstrate the trade-off.

4. The heuristic sentence segmentation may be brittle and it is unclear how the "weights" for the semantic boundaries are instantiated.

5. The experiment task diversity is quite limited as SentKVCompress is only evaluated on a subset of 5 LongBench experiments skewed towards QA, and requires more evidence from additional benchmarks to support its claims. Sentence-level KV approach baselines are also missing.

[1] Zhu, Yuxuan, et al. "SentenceKV: Efficient LLM Inference via Sentence-Level Semantic KV Caching." arXiv preprint arXiv:2504.00970 (2025).

**Questions:**

1. Provide more complete results on a more comprehensive subset of LongBench.
2. Other experiments such as RULER should be run to better support claims.
3. Additional baselines for related sentence-level segmentation approaches should be added.
4. How does the selection process interact with Flash Attention, is it still compatible?
5. Choice and taxonomy of baselines is imprecise. Why is PQCache part of "loading", as it is mainly a KV compression and quantization work? KV quantization (KIVI) does not seem to be related to SentKVCompress, so inclusion in the results does not add much value.

---

> ### Author Response · Authors · 2025-11-30
>
> Hello reviewer, thank you for your comments.
>
> We understand your comments as pertaining to two main aspects: first, the distinctions from related work, and second, matters of detail.
>
> ### 1.The distinctions from related works
>
> As for your comment "Additional baselines for related sentence-level segmentation approaches should be added.": We have selected Sentencekv[1], a representative work at the sentence level, to illustrate the differences. **The most significant difference between our approach and theirs lies in the length control strategy during text split, punctuation choice during text split and the variable-length block processing strategy during sentence selection**. This brings advantages in terms of precision and latency respectively. Next, I will provide a detailed explanation through ablation experiments.
>
> | Accuracy and latency | Triviaqa |  | 2Wikimqa |  | Qasper |  | Average |  |
> |:--------:|:--------:|:--:|:--------:|:--:|:--------:|:--:|:--------:|:--:|
> | Metric | E2e Latency(sec.) | Accuracy(F1 score) | E2e Latency(sec.) | Accuracy(F1 score) | E2e Latency(sec.) | Accuracy(F1 score) | E2e Latency(sec.) | Accuracy(F1 score) |
> | Sentencelkv | 31.07 | 74.89 | 23.75 | 22.64 | 26.09 | 29.08 | 26.97 | 42.20 |
> | ours | 5.49 | 83.10 | 3.65 | 21.87 | 4.05 | 29.59 | 4.40 | 44.85 |
> | **Comparison** | **5.7×** | **+8.21** | **6.5×** | **-0.77** | **6.4×** | **+0.51** | **6.1×** | **+2.65** |
>
> (1) Ablation on length control strategy
>
> | Accuracy: F1 score | Hotpotqa | GovReport | 2WikiMQA | Qasper |
> |--------------------|----------|-----------|----------|--------|
> | Without control    | 24.23    | 30.63     | 13.89    | 19.5   |
> | With control       | 42.01    | 33.15     | 26.26    | 30.2   |
> | **Comparison**         | **+17.78**   | **+2.52**     | **+12.37**   | **+10.7**  |
>
>
> (2) Ablation on punctuation choice strategy
>
> | Accuracy: F1 score | Hotpotqa | GovReport | 2WikiMQA | Qasper |
> |--------------------|----------|-----------|----------|--------|
> | Without choice     | 33.24    | 30.2      | 21.38    | 21.6   |
> | With choice        | 42.01    | 33.15     | 26.26    | 30.2   |
> | **Comparison**         | **+8.77**    | **+2.95**     | **+4.88**    | **+8.6**   |
>
>
> (3) Ablation on variable-length block processing strategy
>
> | Select Latency: ms | Context: 2K | Context: 4K | Context: 8K | Context: 16K | Context: 32K |
> |--------------------|-------------|-------------|-------------|--------------|--------------|
> | Without process    | 6.06        | 11.05       | 27.03       | 53.21        | 105.11       |
> | With process       | 1.63        | 2.36        | 4.43        | 7.50         | 13.75        |
> | **Comparison**         | **3.7x**        | **4.7x**        | **6.1x**        | **7.1x**         | **7.6x**         |
>
>
> All of the ablation experiments mentioned above are included in the current version of the article.
>
>
>
>
>
> ### 2.Matters of detail
> (1) As for your comment " Why is **PQCache** part of "loading", as it is mainly a KV compression and quantization work? KV quantization (KIVI) does not seem to be related to SentKVCompress, so inclusion in the results does not add much value ": PQCache is a work presented at Sigmod 2025, whose core idea is to organize the KVCache as a PQ index during the prefill stage. The PQ index is one of the classic index types in the field of vector retrieval. **During the decode stage, the most relevant KVCache to the query is retrieved from the PQ index and treated as important KVCache, which is then loaded back from the CPU to the GPU for QKV computation in the attention stage.** This method reduces the amount of data loaded, thereby optimizing the memory-bound issue commonly encountered in long-context inference. Therefore, we classify it as a loading-oriented work. We included KV quantization (KIVI) in the comparison because we established a classification framework for KVCache-related work in the PROBLEM SETTING section of the paper, analyzing it from the perspectives of KV Storage and KV Usage. KV quantization falls within this classification framework.
>
>
> (2) As for your comment "it is unclear how the "weights" for the semantic boundaries are instantiated": **We have already provided a detailed presentation of the weights of different punctuation marks under various models in Figure 9 of the appendix in the original paper. **
>
> (3) As for your comment "per-dim min/max key vectors but does not specify..." and " why per-head choices is theoretically or empirically better is needed to demonstrate the trade-off": We have provided a detailed demonstration in Figure 4 (a) of the paper. Taking the attention head dimension of 128 as an example, we take the maximum value across the corresponding text sequence in each dimension, which is referred to as "per max". As for choosing the per-head approach, this is because **it offers higher precision, as the information differs across attention heads.**

---

### Official Review · Reviewer_Qbzv · 2025-11-01

**Soundness:** 2
**Presentation:** 1
**Contribution:** 2
**Rating:** 2
**Confidence:** 4

**Summary:**

This paper proposes SentKVCompress, a sentence-level KV cache compression framework for efficient long-context LLM inference. The authors argue that existing token-level compression methods suffer from either accuracy loss or high computational overhead, and propose to leverage sentence-level semantic structure for both KV storage and usage. The method includes a sentence-aware preprocessing framework and a sentence semantic-driven selection strategy.

**Strengths:**

1. The idea of performing context selection and compression at the sentence level is intuitive and well-motivated.
2. The paper presents both performance and efficiency in long-context processing.

**Weaknesses:**

1. The experimental results only present a very limited subset of tasks from LongBench, making it difficult to assess the method's generalizability.
2. The paper only presents results on LongBench-V1, where the average context length is around 8K tokens. The authors should evaluate on more challenging benchmarks with longer contexts, such as LongBench-V2, RULER.
3. The paper lacks comparisons with several important sparse attention methods, particularly block-sparse attention approaches that are conceptually related to the proposed sentence-level method. Notable missing baselines include MInference, InfLLM, XAttention, and other recent block-based or chunk-based attention mechanisms.

**Questions:**

Please refer to Weaknesses.

---

> ### Author Response · Authors · 2025-11-30
>
> Hello reviewer, thank you for your comments.
>
> In response to your comment, **we have addressed the lack of comparison with block-sparse attention as follows:**
>
> 1.Qualitative comparison
>
> Works like XAttention[1] primarily address the compute-bound problem, which enables their application beyond text to domains like video. In contrast, the memory-bound issue is more critical in long-text scenarios, which is the primary problem we aim to solve. For this reason, we have summarized relevant block-sparse works below and begin with a qualitative comparison. The core approach of InfLLM[2] and Quest[3] involves partitioning the historical KV Cache into fixed-length blocks and compressing each one. During the decode stage, the importance of the Query and each block is computed to pre-select and load critical blocks, thereby reducing the loading overhead for the subsequent QKV computation in the attention phase. The general approach of SentenceKV[4] is similar to the ones mentioned above. The key difference lies in its block split strategy, which splits the text based on punctuation into sentences. This method can better capture textual semantics and improve accuracy. However, the resulting variable-length blocks, due to varying sentence lengths, may impair parallel computing efficiency. **Our work incorporates considerations for both the accuracy of semantic split&Length control and the computational speed of handling variable-length blocks.**
>
> | Related block-sparse attention work | Semantic split | Variable-length block handling strategy | Block Length control |
> | :--- | :--: | ---: | ---: |
> | InfLLM[2] | ×   | × | √ |
> | Quest[3] | ×   | × | √ |
> | Sentencekv[4] | √   | × | × |
> | Ours | √   | √ | √ |
>
> 2.Quantitative comparison
>
> A comparison with the Quest [3] work has been previously mentioned in the original text. Additionally, we have benchmarked our method against the most relevant work, SentenceKV [4], **to highlight our superior speed under comparable accuracy.**
>
> | Accuracy and latency | Triviaqa |  | 2Wikimqa |  | Qasper |  | Result Average |  |
> |:--------:|:--------:|:--:|:--------:|:--:|:--------:|:--:|:--------:|:--:|
> | Metric | E2e Latency(sec.) | Accuracy(F1 score) | E2e Latency(sec.) | Accuracy(F1 score) | E2e Latency(sec.) | Accuracy(F1 score) | E2e Latency(sec.) | Accuracy(F1 score) |
> | Sentencelkv | 31.07 | 74.89 | 23.75 | 22.64 | 26.09 | 29.08 | 26.97 | 42.20 |
> | ours | 5.49 | 83.10 | 3.65 | 21.87 | 4.05 | 29.59 | 4.40 | 44.85 |
> | **Comparison** | **5.7×** | **+8.21** | **6.5×** | **-0.77** | **6.4×** | **+0.51** | **6.1×** | **+2.65** |
>
> **Overall, our additional length-control design ensures optimal text information extraction for accuracy, while the variable-length block strategy maintains computational efficiency for speed.**
>
> [1] Xu R, Xiao G, Huang H, et al. Xattention: Block sparse attention with antidiagonal scoring[J]. arXiv preprint arXiv:2503.16428, 2025.
>
> [2] Xiao C, Zhang P, Han X, et al. Infllm: Training-free long-context extrapolation for llms with an efficient context memory[J]. Advances in Neural Information Processing Systems, 2024, 37: 119638-119661.
>
> [3] Tang J, Zhao Y, Zhu K, et al. Quest: Query-aware sparsity for efficient long-context llm inference[J]. arXiv preprint arXiv:2406.10774, 2024.
>
> [4] Zhu Y, Falahati A, Yang D H, et al. SentenceKV: Efficient LLM Inference via Sentence-Level Semantic KV Caching[J]. arXiv preprint arXiv:2504.00970, 2025.

---

### Note · Authors · 2026-01-27

I have read and agree with the venue's withdrawal policy on behalf of myself and my co-authors.

---

### Meta-Review · Area_Chair_jBoh · 2026-01-13

**Summary:**

The paper proposes SentKVCompress, a sentence-level KV cache preprocessing and selection framework motivated by the claim that token-level KV management is inherently “unstructured” and leads to either accuracy loss or high overhead. Across reviews, the dominant concerns are that the contribution is incremental relative to very recent sentence/chunk-level KV work (especially SentenceKV) and that the empirical support is not strong enough for the paper’s scope claims: multiple reviewers emphasize missing evaluations on genuinely long-context settings (e.g., RULER, LongBench-v2, or 32k–128k style regimes) and limited task diversity (a small subset of LongBench skewed toward QA). Reviewers also criticize baseline coverage and positioning against closely related block/chunk-based inference methods (e.g., InfLLM/Quest-style block approaches and other sparse/block attention systems) and note that the methodology description is under-specified in key places (how sentence statistics are used in scoring/selection; why per-head iteration is necessary despite breaking parallelism; how punctuation weights are set and how the splitter behaves on punctuation-sparse inputs like math/code). Given three rejects at score 2 and one marginally-below-threshold 4, the review signal is clearly negative, and the paper does not have the kind of rebuttal-driven reviewer flips that would justify acceptance under a selective bar, so the AC recommendation is Reject.

**Reviewer Concerns:**

The author response usefully clarifies several details and adds material that partially addresses novelty/positioning concerns: they provide a qualitative taxonomy versus block-based methods and a quantitative comparison versus SentenceKV, along with ablations for length control, punctuation choice, and variable-length block processing; they also give additional statistics on chunk length distribution and claim mean-pooled sentence representations underperform their per-dimension extrema approach. However, the response does not address the most repeatedly requested validation gap—evaluation on truly long-context benchmarks such as RULER / LongBench-v2 or stronger context-length stress tests—nor does it add several of the key missing baselines that reviewers asked for (or a compelling substitute evaluation that would make the omission acceptable). Importantly, there is no evidence in the record that reviewers updated their scores after the response, so it would be inappropriate to infer meaningful score movement.

**Reviewer Scores:**

Qbzv (initial 2) did not participate after the author response; their main asks were broader task coverage and longer-context benchmarks (RULER/LongBench-v2) plus missing block-sparse/chunk baselines, none of which were fully satisfied, so the expected post-rebuttal score is 2.

7Vi3 (initial 2) did not participate after rebuttal; while the authors added a SentenceKV comparison and extra ablations, their remaining core concerns (limited benchmark diversity, missing long-context evaluation, and under-specified method details and tradeoffs around per-head processing) remain, so the expected post-rebuttal score is 2.

hX5P (initial 2) did not participate after rebuttal; the response addresses SentenceKV comparison and provides some additional analysis, but does not add the long-context benchmark evidence they explicitly called “essential,” so the expected post-rebuttal score is 2.

Xhab (initial 4) did not participate after rebuttal; the response partially addresses novelty and adds the SentenceKV comparison, but the missing long-context benchmarks and broader baselines remain outstanding, so the expected post-rebuttal score is 4.

---

### Decision · Program_Chairs · 2026-01-26

Reject